# A Review of the Application of Modified Separators in Inhibiting the “shuttle effect” of Lithium–Sulfur Batteries

**DOI:** 10.3390/membranes12080790

**Published:** 2022-08-17

**Authors:** Bo-Wen Zhang, Bo Sun, Pei Fu, Feng Liu, Chen Zhu, Bao-Ming Xu, Yong Pan, Chi Chen

**Affiliations:** 1Hubei Provincial Key Laboratory of Green Materials for Light Industry, Hubei University of Technology, Wuhan 430068, China; 2Shandong Zhongsheng Pharmaceutical Equipment Co., Ltd., Yantai 264010, China

**Keywords:** lithium–sulfur batteries, polysulfide, “shuttle effect”, separator, modification

## Abstract

Lithium-sulfur batteries with high theoretical specific capacity and high energy density are considered to be one of the most promising energy storage devices. However, the “shuttle effect” caused by the soluble polysulphide intermediates migrating back and forth between the positive and negative electrodes significantly reduces the active substance content of the battery and hinders the commercial applications of lithium–sulfur batteries. The separator being far from the electrochemical reaction interface and in close contact with the electrode poses an important barrier to polysulfide shuttle. Therefore, the electrochemical performance including coulombic efficiency and cycle stability of lithium–sulfur batteries can be effectively improved by rationally designing the separator. In this paper, the research progress of the modification of lithium–sulfur battery separators is reviewed from the perspectives of adsorption effect, electrostatic effect, and steric hindrance effect, and a novel modification of the lithium–sulfur battery separator is prospected.

## 1. Introduction

Along with the development of mobile electronics and electric vehicles, a trend to develop energy storage devices with high efficiency and high specific energy has emerged [1,2,3]. Rechargeable batteries with long service life and high energy storage efficiency are attracting much attention from researchers. Among them, lithium–sulfur batteries (LSBs) have high theoretical specific capacity (1675 mAh g^−1^) and high energy density (2600 Wh kg^−1^) [4,5], and the cathode sulfur is low cost, abundant, and environmentally friendly. Therefore, LSBs have great development prospects [6]. The charge–discharge process of LSBs is a dissolution–deposition reaction [7]. During the discharge process, the cathode sulfur (S_8_) is electrochemically reduced to soluble long-chain lithium polysulfide Li_2_S_x_ (4 ≤ x ≤ 8) first, and then converted to insoluble short-chain Li_2_S_2_/Li_2_S. The charging reaction is the opposite of the discharging reaction. The solid Li_2_S_2_/Li_2_S is first converted to short-chain LiPS, which is then further oxidized to soluble long-chain Li_2_S_x_ (4 ≤ x ≤ 8), and finally to solid S_8_. The reaction is as follows [8]:Negative electrode: 16 Li ⇌ 16 Li^+^ + 16 e^−^(1)
Positive electrode: S_8_ + 16 Li^+^ + 16 e^−^ ⇌ 8 Li_2_S(2)

The “shuttle effect” of LSBs is known to be an important factor limiting their practical application [9,10,11,12]. The “shuttle effect” refers to the phenomenon that Li_2_S_x_ (4 ≤ x ≤ 8) produced by the positive electrode diffuses to the negative electrode during the charging and discharging process, and is reduced to solid Li_2_S_2_/Li_2_S on the negative electrode surface and attached to the negative electrode. It can cause irreversible loss of battery active material, increase battery internal resistance, and decrease the theoretical capacity of LSBs [13,14,15]. At present, researchers have proposed many methods to suppress the “shuttle effect” of LSBs, such as electrode modification, electrolyte optimization, and separator modification. Among them, the separator is far away from the electrochemical reaction interface and in close contact with the electrode, and more importantly, it is also the diffusion channel of polysulfides (LiPSs). Therefore, its structure and performance greatly affect the electrode reaction of LSBs and the shuttle of LiPSs between the two electrodes. However, currently, the most commercialized and widely used polyolefin separators (such as polyethylene (PE) and polypropylene (PP)) have large pore sizes and do not interact with LiPSs, so they cannot inhibit the shuttle of LiPSs between the two electrodes of LSBs [16,17]. Therefore, under the premise of ensuring the basic functions of the polyolefin-based separator, if it is modified to have the function of adsorbing and/or converting LiPSs, it will be able to inhibit the “shuttle effect” of LiPSs. [18,19,20]. From what is known in the literature [21,22,23,24], the current separator modifications of LSBs are mainly based on confining LiPSs to the cathode region of the battery during their migration to the anode. This can be done from three perspectives: (1) modification by adsorption effect, making the separator adsorbable and fixing LiPSs on the positive side; (2) modification by electrostatic effect, making the separator repulsive and inhibiting the migration of LiPSs; (3) modification by positioning barrier effect, reducing the pore size of the separator and keeping the LiPSs in the positive electrode region by means of physical barrier. These three methods are described below.

## 2. Adsorption Effect Modification

### 2.1. Physical/Chemical Adsorption

The modification of the separator of LSBs by means of the physical adsorption effect generally refers to the use of the van der Waals force between LiPSs and the trapping material to capture LiPSs. Separator materials with high porosity and large specific surface area can provide more trapping sites, which are beneficial to inhibit the migration of LiPSs between the electrodes [25]. Therefore, attaching porous carbon materials such as carbon nanofibers, carbon flakes, microporous carbon nanofibers, reduced graphene oxide, etc. as a coating to the separator has been widely used in the separator modification of LSBs. The modification effects of different carbon materials are shown in Table 1. Generally speaking, the high sulfur loading of the cathode will increase more LiPSs in the electrolyte and result in a severer shuttle effect of LSBs. From Table 1, it can be seen that the improvement of using light mesoporous carbon as the coating to modify the separator is the most significant. At the high sulfur loading (3.5 mg cm^−2^), the LSBs with this separator have a mass specific capacity of 1021 mAh g^−1^ after 100 cycles at 0.5 C, with a capacity decay rate of only 0.081%. This may be because the large number of mesopores (12 nm) in the mesoporous carbon can tune the huge volume change of S_8_ during the lithiation process, thereby suppressing the loss of polysulfides, which in turn controls the shuttle of LiPSs and improves the electrochemical performance of LSBs.

Besides carbon materials, non-carbon materials can also be used as coatings to enhance the adsorption capacity of the separators of LiPSs. Deng et al. [16] prepared a polymetaphenylene isophthalamide (PMIN) separator by electrospinning. The addition of tetrabutylammonium chloride to the spinning solution produces a separator with high porosity. Using the dendritic nanofiber separator with good physical trapping effect, the LSBs exhibited a capacity decay rate of only 0.049% after 800 cycles at 0.5 C. 

Chiu et al. [36] modified the PP separator of LSBs with a mixture of ethylene oxide (PEO) and lithium bis(trifluoromethanesulfonyl)imide (LiTFSI) as a coating. The experimental results show that the lithium-ion diffusion coefficient (9.6 × 10^−9^~3.0 × 10^−8^ cm^2^ s^−1^) of PEO/LiTFSI coating is significantly higher than that of pure PP separator (7.6 × 10^−9^~2.2 × 10^−8^ cm^2^ s^−1^). The LSBs with PEO/LiTFSI separator achieve an initial discharge capacity of 1212 mAh g^−1^ at 0.1 C and can still maintain a high reversible capacity of 534 mAh g^−1^ and a stable coulombic capacity after 200 cycles. This may be due to the fact that PEO can inhibit the diffusion of LiPSs as a gel polymer electrolyte, and the addition of LiTFSI salt enhances the ability of the lithium ion transfer of the PEO coating. The electrochemical efficiency and the synergistic effect of the PEO/LiTFSI coating are shown in Figure 1.

Although the modification of polyolefin separators with carbon materials can inhibit the shuttle of LiPSs between the two electrodes of LSBs to a certain extent, the interaction force between non-polar carbon materials and polar LiPSs is weak. Therefore, inhibiting the diffusion of LiPSs in the LSBs only by physical adsorption is not satisfied [37]. Compared with physical adsorption, chemical adsorption, which relies on the formation of chemical bonding forces between LiPSs and the surface atoms of adsorbent materials, can make a higher selectivity and achieve a better immobilizing effect of LiPSs [38]. For example, doping electronegative heteroatoms (such as N, O, and S, etc.) in carbon coatings can help trap LiPSs by the formation of Li–X bonds (dipole–dipole interactions) between heteroatoms and LiPSs [39,40]. Hou et al. [41] analyzed the ability of various atomically doped graphene nanoribbons (GNRs) to adsorb sulfur and Li_2_S_x_ by density functional theory (DFT). It was found that the adsorption effect of N and O (binding energy −2.53~−2.56 eV)-doped GNRs on Li_2_S_4_ is much greater than that of B, F, S, P, and Cl (binding energy −1.93 eV), which is due to the existence of dipole–dipole electrostatic interactions (~1.95 eV). The relationship between element electronegativity and binding energy is shown in Figure 2.

Zeng et al. [42] designed a honeycomb-like N, P diatomic doped carbon (HNPC) modified separator, in which the HNPC coating can effectively anchor the LiPSs by forming N–Li and P–S bonds. When using the acetylene black-sulfur composite as the cathode with the sulfur content of 79.7%, the LSBs showed excellent long-cycle stability with a capacity decay rate of only 0.06% per cycle after 900 cycles at 1 C. Figure 3 shows the cycle performance of separators modified by different coatings. It can be seen that the chemical confinement of LiPSs by diatomic doping is higher than that of single-atom doping systems.

Besides polar atoms, the doping of polar functional groups can also realize the chemisorption of the separator to LiPSs. Pei et al. [43] used polydopamine/polyethylene oxide (PDA/PEI) to modify polyolefin separators and achieved desired results. The PDA/PEI-modified separator made the initial discharge capacity of LSBs 1250 mAh g^−1^ at 0.2 C and it remained900 mAh g^−1^ after 100 cycles. This could be attributed to the formation of numerous C=N bonds between the amine groups in PEI and dopamine, which inhibited the accumulation of dopamine oligomers on the separator through hydrogen bonds or π−π bonds, thereby making the separator hydrophilic, and the enriched N and O functional groups of the PDA/PEI coating effectively adsorbed LiPSs in the electrolyte. 

In addition, the modification of PP separator with metal materials, metal oxides, and metal sulfides, the exposed metal (M) sites of which can form S–M bonds with LiPSs, are also beneficial to improve its adsorption capacity to LiPSs [44]. Tao et al. [12] calculated by DFT that the binding energy of metal oxides and LiPSs (from −1.54 to −7.12 eV), is higher than that of heteroatoms and LiPSs (from −2.53 to −2.56 eV). Liu et al. [45] designed graphene oxide (GO) coatings doped with TiO_2_ nanoparticles, as shown in Figure 4a. The Ti–O–C bond formed between TiO_2_ and GO and the wrinkled sheet structure of GO make the combination of TiO_2_ and GO tight, which helps to enhance the electrical conductivity of the separator. Meanwhile, the S–Ti–O and Ti–S bonds formed between TiO_2_ and S enhanced the ability of the separator to adsorb LiPSs. The LSBs coated with TiO_2_/GO separator can retain a reversible capacity of 843.4 mAh g^−1^ after 100 cycles at 0.2 C (Figure 4b).

### 2.2. Catalytic Adsorption

Physical/chemical adsorption can suppress the “shuttle effect” of LiPSs, but under the conditions of high sulfur loading (>5 mg cm^−2^) or long cycling, the increase of LiPSs dissolved in the electrolyte will lead to slow reaction kinetics. Moreover, the limited adsorption sites on the separator are not enough to adsorb LiPSs in the electrolyte fully, thus resulting in a low specific capacity and short cycle life of the battery. Therefore, the ability of the separator to suppress the “shuttle effect” may be reduced or even lost [46,47]. The introduction of catalytic substances on the surface of the separator can not only reduce the energy barrier of the conversion of LiPSs and insoluble Li_2_S_2_/Li_2_S, accelerating the electrochemical reaction of LSBs, but also transfer the adsorbed LiPSs to the redox reaction in time. Thus, the content of the active substances of the battery is kept to the greatest extent, and the cycle performance of LSB is significantly improved [14].

Inherent defects in the metal oxide structure can provide more active sites for trapping and transforming LiPSs [48]. Lv et al. [49] loaded bimetallic NiCo_2_O_4_ nanoparticles onto reduced graphene oxide as a catalytic coating for the separator. During the electrode reaction process, the oxidized metal ions (Ni^3+^ and Co^3+^) combine with LiPSs to form Ni–S and Co–S bonds, which anchor the LiPSs to the NiCo_2_O_4_@rGO surface to suppress the “shuttle effect” of LBSs. Furthermore, DFT calculations show that the NiCo_2_O_4_ (100) crystal surface has a low Li-ion diffusion barrier (0.15 eV compared to 0.293 eV for carbon materials), which helps to improve the migration rate of lithium ions and promote lithium-ion–electron coupling, thereby accelerating the conversion of LiPSs to Li_2_S_2_/Li_2_S. Using a core-shell structure material (encapsulated sulfur in carbon) with a sulfur content of 70 wt% as the cathode and under the condition of a high sulfur loading of 6 mg cm^−2^, the LBSs showed the capacity loss rate only 0.02% after 400 cycles at a current density of 1 mA cm^−2^, and at the same time, there was an excellent areal capacity of 7.1 mAh cm^−2^.

Adding other metals to monometallic compounds can also improve the reaction kinetics of LiPS transformation on the surface of metal compounds [50]. Zhang et al. [51] introduced Mo atoms into Ni_3_N and used a composite of multi-walled carbon nanotubes and Ni_0.2_Mo_0.8_N to modify the separator. During the electrochemical reaction process, the Mo atoms in the Ni_0.2_Mo_0.8_N are etched and leached out by LiPSs, leaving a large number of vacancies around Ni (Figure 5a). The vacancies formed can accelerate the charge transfer and the LiPS conversion. The potential of the Ni_3_N and Mo_2_N surfaces in the coating is higher than lithium negative electrode, thus creating an electric field between the separator and the lithium anode (Figure 5b), which promotes the directional movement of sulfur and lithium ions and inhibits the shuttling of LiPSs to some extent. The LSBs with this separator can achieve an initial battery capacity of 1097.2 mAh g^−1^ at a high rate of 5 C.

Due to the abundant pore structures and catalytic sites, metal–organic frameworks (MOFs) composed of organic ligands and transition metal ions have been widely used for catalytic modification of the separators of LBSs. The pore structure in MOFs can promote the adequate contact between the separator and the electrolyte and limits the diffusion of LiPSs, while the metal central ions can chemisorb and catalyze LiPSs [41]. Hong et al. [52] doped cerium-based metal-organic frameworks (Ce-MOFs) into CNTs to form Ce-MOFs/CNT separator coatings. The large specific surface area of Ce-MOFs and the ligated unsaturated Ce (IV) cluster nodes can rapidly adsorb LiPSs and accelerate the conversion between LiPSs and Li_2_S_2_/Li_2_S, which is shown in Figure 6a. As seen, under the high sulfur loading of 6 mg cm^−2^, the initial specific capacity of LSBs was 1021.8 mAh g^−1^ at 1 C, which slowly decreased to 838.8 mAh g^−1^ after 800 cycles with a decay rate of only 0.022% per cycle, as shown in Figure 6b. The coulombic efficiency of the LBSs was close to 100%, which suggesting its excellent cycle performance.

In addition to transition metal elements, rare-earth elements have also been used in the catalytic modification of the separators of LBSs. Peng et al. [53] prepared a novel pp separator coated with Eu_2_O_3_/Ketjen black (Eu_2_O_3_/KB). The exposed (222) plane of Eu_2_O_3_ exhibits strong polarity-polarity interaction with LiPSs, which is beneficial to confine the transfer of LiPSs. The adsorption energy of Eu_2_O_3_ with LiPSs is shown in Figure 7. Apart from that, oxygen vacancies in Eu_2_O_3_ can provide more catalytic sites for regulating Li_2_S precipitation. Therefore, with high crystal face index, Eu_2_O_3_ can lower the energy barrier for Li_2_S nucleation on it and promote the conversion of LiPSs to Li_2_S. The LBSs with Eu_2_O_3_/KB modified separator exhibited excellent cycling stability and rate capability. Its capacity decay rate achieved only 0.05% per cycle during 500 cycles at 1 C.

Some non-metallic materials with catalytic activity have also been used in the separator of LBSs to promote the conversion of LiPSs. For example, Wang et al. [54] used red phosphorus (RP) as a coating to enhance the sulfur reaction kinetics of LSBs. Through Lewis acid–base interaction, The Li_3_PO_4_ with high conductivity was formed by RP and LiPSs, which can improve the conductivity of lithium ions and promote the redox reaction of sulfur. However, the formation of Li_3_PO_4_ consumes some active sites for adsorbing LiPSs in the separator, which may lead to the decrease of the capacity of the LSBs under long-term cycling. The LSBs with RP-modified separator exhibited a capacity of 729.6 mAh g^−1^ after 500 cycles at 1 C with a capacity retention rate of 82%.

## 3. Electrostatic Effect

The sulfur atoms at the end of LiPSs possess a large number of negative charges and free radicals. The introduction of negatively charged species on the separator can inhibit the shuttle of polysulfide ions through electrostatic repulsion and simultaneously facilitate Li^+^ migration. This could decrease the resistance between the positive electrode and the separator, preventing the energy generated by the battery from dissipating in the form of thermal energy [39,55].

### 3.1. Negatively Charged Group

Song et al. [56] grafted polyacrylic acid (PAA) on the surface of PP separator (PP-g-PAA). A large number of carboxyl functional groups in PAA can not only repel LiPSs anions through electrostatic interaction, but also enhance the hydrophilicity of the separator and reduce the internal resistance of LSBs. Chiu et al. [57] prepared a gel polymer electrolyte separator coating by mixing polymethyl methacrylate with rich -COO- and LiTFSI. When the sulfur loading of the cathode was increased from 4 to 10 mg cm^−2^, the lithium-ion diffusion coefficient of LSBs slightly decreased from 5.27 × 10^−9^–3.1 × 10^−8^ cm^2^ s^−1^ to 2.09 × 10^−9^–1.27 × 10^−8^ cm^2^ s^−1^, which indicates that the coating enhances the lithium-ion transfer efficiency and suppresses the diffusion of LiPSs. Under the high sulfur loading of 8 mg cm^−2^, LSBs with this separator exhibit a high area capacity of 7.1 mWh cm^−2^. Li et al. [58] modified PP with Fe_3_O_4_ nanoparticles and carboxylated carbon nanotubes (CTs-COOH). The Fe_3_O_4_ layer acts as an “upper current collector” and facilitates electron transport, while -COOH can inhibit the transport of LiPSs through electrostatic effects. The synergistic effect of the two factors enables LSBs to have a capacity decay rate as low as 0.032% per cycle when cycled for 2000 times at 1 C.

Nafion with rich sulfonic acid groups (-SO_3_H) is also widely used in the modification of the separators of LSBs. But due to the poor conductivity, an overly thick coating of Nafion will reduce the lithium-ion transport capacity of the separator [59]. Zhuang et al. [60] designed a ternary-layer separator composed of PP, graphene oxide (GO) and Nafion for LSBs (Figure 8), in which the ultrathin GO layer (0.0032 mg cm^−2^, about 40 layers) can cover the macropores of the PP matrix and provides support for the Nafion layer. The Nafion layer with the areal loading of 0.05 mg cm^−2^ also retains the conductivity of the separator while repelling LiPSs. The specific capacity of LSBs using the ternary layer separator can reach a discharge capacity of 1057 mAh g^−1^ at 0.1 C.

In addition, the confinement of LiPSs can also be achieved by using the charge environment between the two-dimensional material layers. Xu et al. [61] used two-dimensional vermiculite nanosheets to prepare the separators of LSBs, the structure of which is shown in Figure 9. The vermiculite nanosheet is negatively charged, so the positively charged lithium ions can be transferred between the layers, while the negatively charged polysulfide ions cannot diffuse due to electrostatic interaction. This effectively suppressed the transmembrane diffusion of LiPSs.

### 3.2. Polar Particles

Besides negatively charged groups, some polar particles can also repel LiPSs through electrostatic effects. BaTiO_3_ (BTO) with high dielectric constant can generate a large number of positive and negative charges on the surface of BTO through spontaneous polarization, which can electrostatically interact with lithium ions and polar molecules LiPSs [62]. Yim et al. [63] embedded BTO particles into PE separators. The polar BTO forms permanent dipoles under the action of electric field (Figure 10), which can block LiPSs through electrostatic repulsion, thereby improving the cycling performance of LSBs. The polarized BTO coating can also enhance the mechanical stability of the PE separator and improve the capacity and cycle life of the LSBs. At a current density of 0.5 C, the LSBs with this separator applied achieves an initial discharge capacity of 1122.1 mAh g^−1^.

## 4. Steric Hindrance Effect

The pore size of commonly used polyolefin separators (about 100 nm) is much larger than the size of LiPSs, which cannot block the shuttle of LiPSs between the cathode and anode of LSBs. Thus, reducing the pore size of the diaphragm can hinder the diffusion of LiPSs to the anode [64]. The modification methods of LSBs separators based on this steric hindrance effect can be broadly classified into two categories: (1) accumulation of particles on the surface of the separator to form a densely modified layer covering the original macropores; (2) construction of a new microporous separator using microporous materials.

### 4.1. Particle Buildup

Maletti et al. [65] prepared a lithium vanadium oxide (LiV_3_O_8_)-coated separator. The dense coating formed by LiV_3_O_8_ nanoparticles can cover the original macropore structure of the PP separator, forming a physical barrier to inhibit the diffusion of LiPSs. LiV_3_O_8_ can disproportionate with LiPSs to promote the conversion of LiPSs to Li_2_S_2_/Li_2_S (Figure 11). In addition, the LiV_3_O_8_ coating significantly reduces the resistance of the cell by facilitating charge transfer at the cathode/diaphragm coating interface. Compared with pure PP, the charge transfer resistance of LSBs with LiV_3_O_8_ separator was significantly reduced by 83%, with a high initial discharge capacity of 1254 mAh g^−1^ at a current density of 0.2 C and a capacity decay rate of 0.063% after 500 cycles at 0.5 C.

Besides the pore size reduction that is possible by loading zero-dimensional nanoparticles on the surface of the separator, the layered structure formed by the stacking of two-dimensional nanosheets can also form a physical barrier to intercept LiPSs. Ghazi et al. [66] deposited multilayer MoS_2_ nanosheets with a thickness of about 350 nm on the surface of polyolefin separators. The formed dense structure can effectively block LiPSs, and at the same time, the voids between the MoS_2_ nanosheets can prevent solid LiS from forming an insulating layer on the separator and provide a transfer channel for lithium ions, which realizes the effective sieving of LiPSs and lithium ions.

Chang et al. [67] designed a two-dimensional MOF sheet molecular sieve (CuBDCs) coating with one-dimensional vertical channels. The structure is shown in Figure 12. The one-dimensional channels (about 5.2 Å) in the CuBDCs nanosheets are much smaller than the size of polysulfides, which can act as molecular sieves and hinder the diffusion of LiPSs to the anode. Secondly, the CuBDC sheet is approximately 10 nm thick, which combined with its internal one-dimensional channels shortens the ion transport path, improving the mobility of Li ions. The LSBs using this separator can still maintain a specific capacity of 830 mAh g^−1^ after 1000 long cycles at 1 C. This indicates that the 2D CuBDC sheets have excellent sieving ability for LiPSs.

### 4.2. New Microporous Separator

A commonly used way to reduce the pore size of polyolefin separators is to coat their surfaces with small-pore-size coatings, and carbon coatings with nanopores have been shown to have good performance in inhibiting the “shuttle effect” of LiPSs under high sulfur-loading conditions [68]. However, the surface energy of the polyolefin separator is low, and it is difficult to adhere a uniform coating material on the surface of the separator. Therefore, the phenomenon of the coating peeling off the separator during the cycle occurs occasionally. Using novel functional separators as substrates is one of the fundamental solutions to this problem.

Compared with the traditional polyolefin separator, the electrospun nanofiber separator has higher porosity and chemical stability. The strong affinity of electrospun nanofibres for the electrolyte can significantly improve the mobility of lithium ions in LSBs, thereby effectively inhibiting the “shuttle effect” of LSBs [69]. Pei et al. [70] prepared a structurally asymmetric aramid nanofiber Janus separator (ANF-JS). The separator structure is shown in Figure 13. The side of the separator facing the positive electrode is a nanoporous layer with ion selectivity, which can hinder the shuttling of LiPSs between electrodes; the side of the separator facing the negative electrode is a microporous layer with a three-dimensional porous structure, which can provide channel for electrolyte absorption and fast lithium-ion transport. After 1000 cycles at 1 C with this separator, the capacity decay rate of the LSBs is reduced to 0.0195% per cycle.

Doping functional particles in the novel separator can combine the steric hindrance effect with other effects, enabling the separator to further inhibit the shuttle of LiPSs [71]. Guo et al. [72] prepared a flexible Ti_4_O_7_/C nanofiber barrier layer (TCNFs) by electrospinning. Figure 14a shows the adsorption–desorption and BET curves, and it can be seen that the pore size of the separator was between 2 and 16 nm and the specific surface area was 40 m^2^ g^−1^, which can effectively isolate LiPSs form the positive side of LSBs. The TCNFs separator has dual effects of physical shielding and chemical adsorption on LiPSs, which make the capacity fade of the LSBs only 0.03% per cycle after 2500 cycles, even at a current density of 3 C, as shown in Figure 14b.

Wang et al. [73] prepared separators (ZIF-7@PCF) with cross-linked structures by grewing two-dimensional MOF sheets (ZIF-7) on porous carbon nanofibers (PCFs), as shown in Figure 15a. The pores of the separator are mostly mesopores and micropores with diameters of less than 10 nm, which are beneficial to hinder the transport of LiPSs. PCF porous carbonaceous framework can facilitate electron and ion transport in batteries, while ZIF-7 MOF provides an abundant active site for the capture and conversion of LiPS. Under high sulfur-loading conditions of 6 mg cm^−2^, the LSB with ZIF-7@PCF exhibits an initial discharge specific capacity of 1221 mAh g^−1^ at a current density of 0.2 C and a discharge specific capacity of 661.3 mAh g^−1^ at 5 C, showing good rate performance, as shown in Figure 15b.

## 5. Conclusions

This paper mainly introduces the methods of modifying the lithium–sulfur battery separators by the adsorption effect, electrostatic effect, and steric hindrance effect using polyolefin separator as the matrix. Compared with the traditional polyolefin separator, the modified separators have good function for inhibiting the “shuttle effect” of LiPSs. Especially, the modified separator with catalytic effect can maximize the utilization of the active species in LSBs, showing excellent cycling and rate performance under long-cycle conditions and high sulfur loading. The current separator modification materials for high sulfur loading and low electrolyte LSBs are mainly carbon-based composites with hollow or layered structures, and the robust internal porous network structure can provide buffers for sulfur volume changes and the shuttle of LiPSs. However, overly thick carbon coating will reduce the actual energy density of the battery, so lightweight materials such as graphene can be selected as the base layer to reduce the thickness of the coating and the load quality. To realize the high energy density and rate capability of LSBs, it is possible to consider using nanoscale catalytic materials with large specific surface area and short ion transport distance or gel electrolyte materials with high ionic conductivity to modify the separator. In addition, the functional modification of novel nanofiber separators with high chemical stability and electrolyte affinity also opens up new avenues for developing high-energy-density LSB separators.

## Figures and Tables

**Figure 1 membranes-12-00790-f001:**
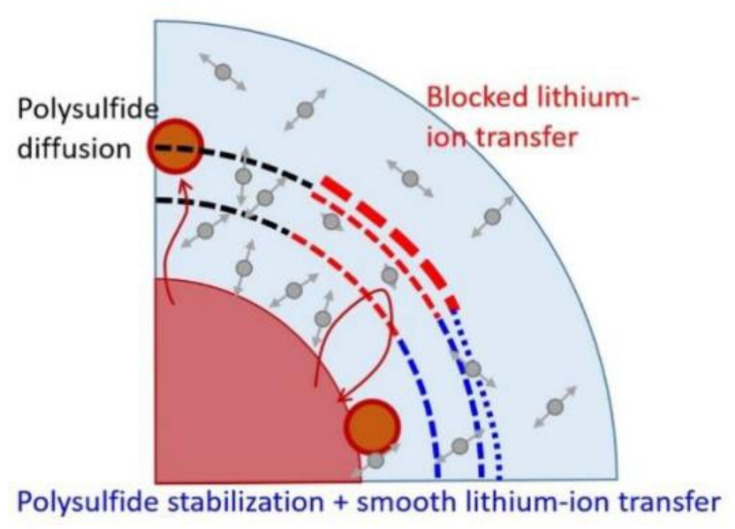
Schematic diagram of PEO/LiTFSI coating hindering LiPSs [36].

**Figure 2 membranes-12-00790-f002:**
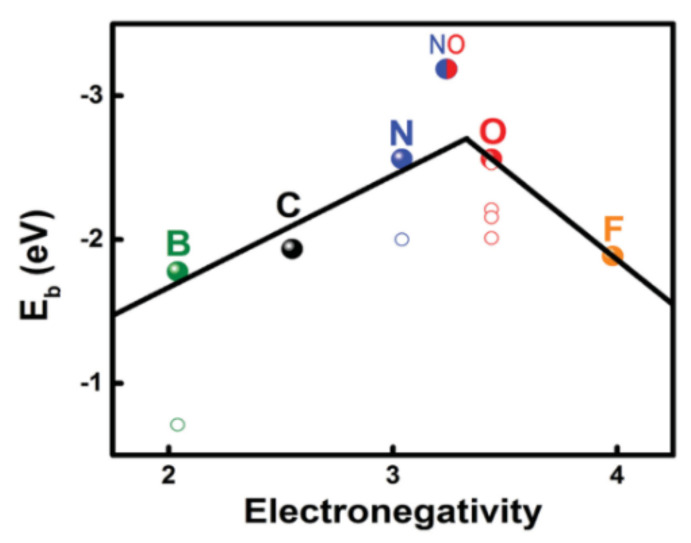
The relationship between the binding energy (E_b_) of the doping element−Li_2_S_4_ and the electronegativity of the element [41].

**Figure 3 membranes-12-00790-f003:**
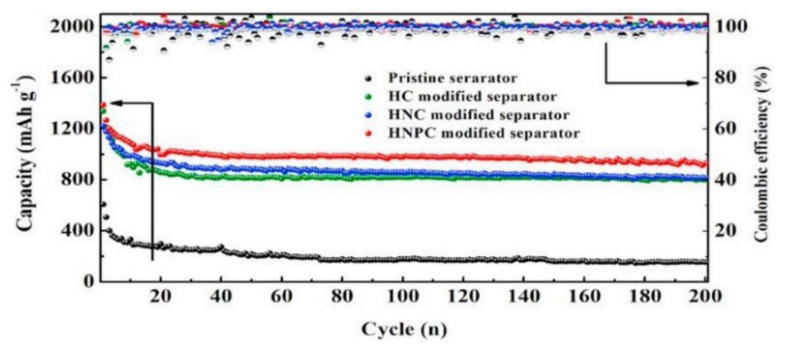
Cycle performance of LSBs using different separators at 0.2 C [42].

**Figure 4 membranes-12-00790-f004:**
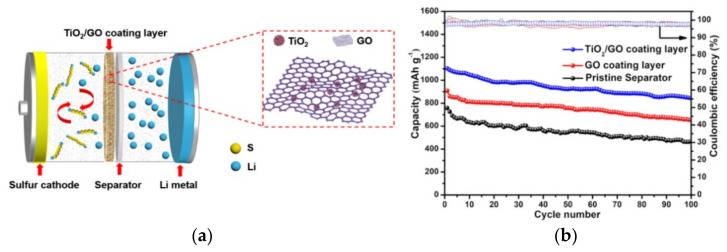
(**a**) Schematic diagram of LBSs with TiO_2_/GO coated functional separator. (**b**) Cyclic performance and coulombic efficiency of the LSBs batteries with different coated separators at a rate of 0.2 C [45].

**Figure 5 membranes-12-00790-f005:**
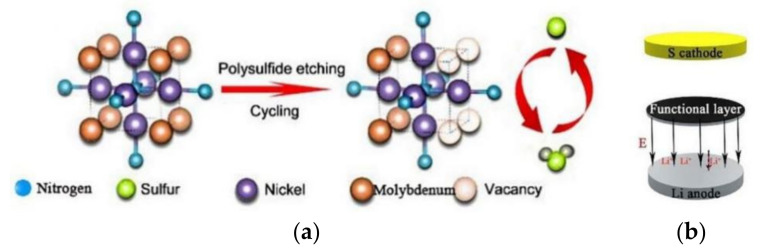
(**a**) Mechanism of in-situ etching of LiPSs by bimetallic nitride. (**b**) Direction of the electric field inside the LSBs [51].

**Figure 6 membranes-12-00790-f006:**
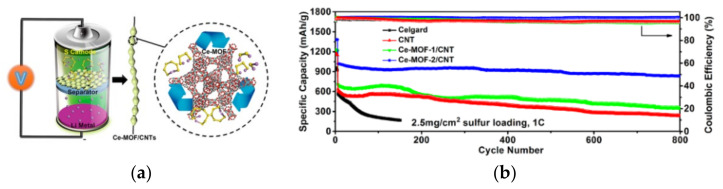
(**a**) Schematic representation of the catalytic conversion of LiPSs on the surface of Ce-MOFs/CNT separators. (**b**) Cyclic performance of cells with different separator at 1 C for 800 cycles (2.5 mg/cm^2^ sulfur loading) [52].

**Figure 7 membranes-12-00790-f007:**
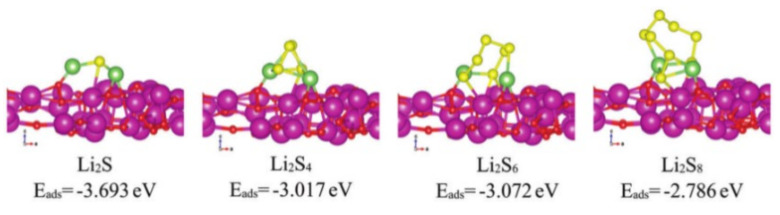
Adsorption energies of different LiPSs on the Eu_2_O_3_ (222) plane [53].

**Figure 8 membranes-12-00790-f008:**
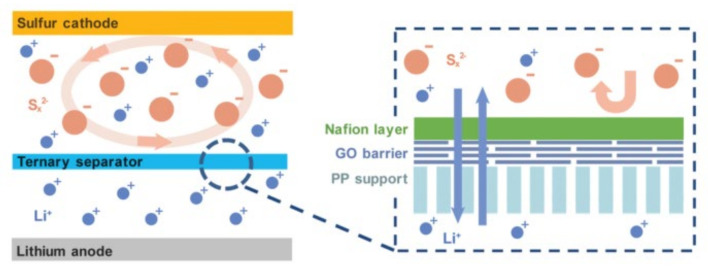
Schematic diagram of ternary PP/GO/Nafion separator [60].

**Figure 9 membranes-12-00790-f009:**
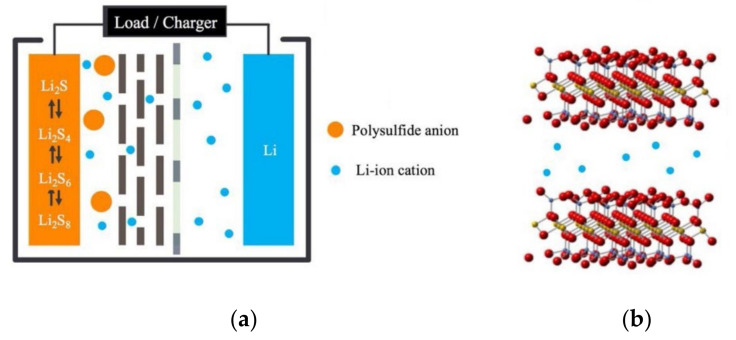
(**a**) Schematic diagram of LSBs sheet vermiculite separator, (**b**) vermiculite sheet structure [61].

**Figure 10 membranes-12-00790-f010:**
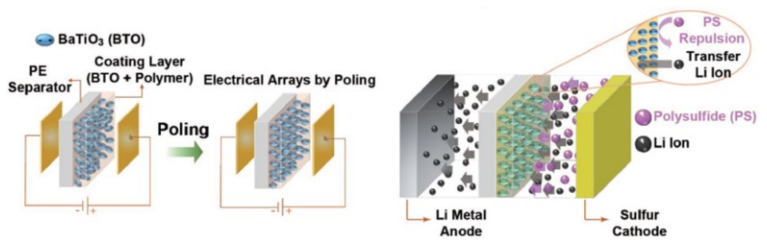
Schematic diagram of the polarization process of BTO/PE and the repulsive effect of polarized BTO on LiPSs [63].

**Figure 11 membranes-12-00790-f011:**
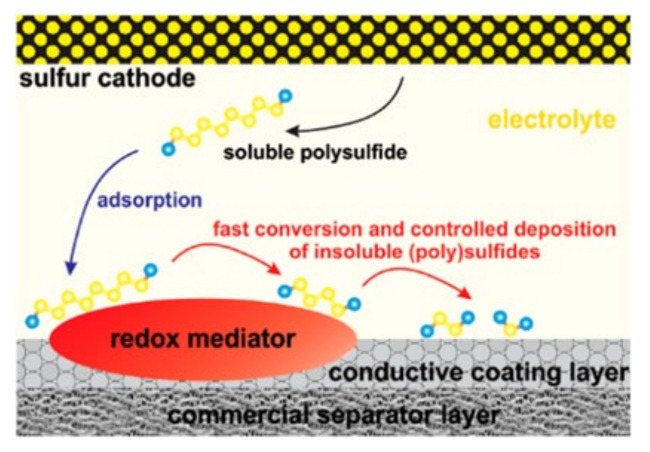
Schematic representation of the adsorption and redox transformation of LiPSs at the LiV_3_O_8_ diaphragm interface [65].

**Figure 12 membranes-12-00790-f012:**
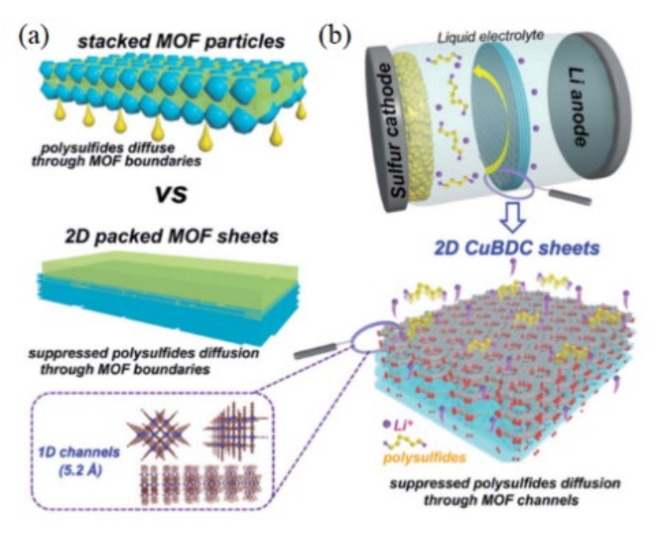
(**a**) Schematic illustration of granular and 2D sheet MOFs for inhibiting LiPSs; (**b**) schematic illustration of 2D CuBDC sheets with crystal structures of 1D nanochannels and 2D CuBDC sheets [66].

**Figure 13 membranes-12-00790-f013:**
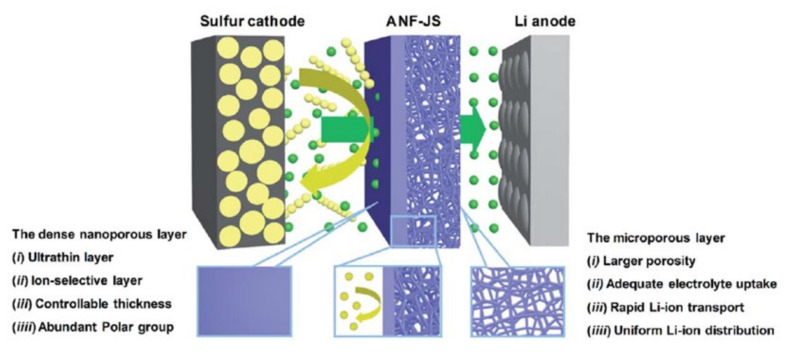
Schematic illustration of ANF-JS applied to LiPSs and Li-ion transport in LSBs [70].

**Figure 14 membranes-12-00790-f014:**
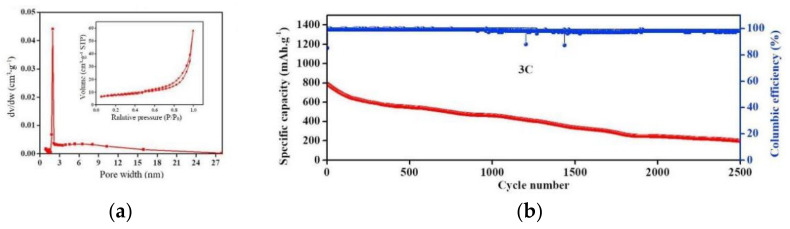
(**a**) Pore size distribution and N_2_ adsorption−desorption isotherms of TCNFs; (**b**) cycling performance of CMK3−S/TCNFs at high current density 3 C [72].

**Figure 15 membranes-12-00790-f015:**
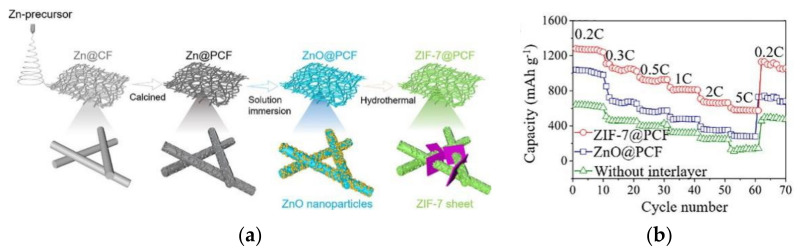
(**a**) Schematic diagram of preparation process of ZIF-7@PCF interlayer; (**b**) the rate capability of cells with ZIF-7@PCF, ZnO@PCF interlayer, and without interlayer [73].

**Table 1 membranes-12-00790-t001:** The properties of PP separators modified with different carbon materials in LSBs.

Materials	Total Pore Volume/cm^3^ g^−1^	Surface Area/m^2^ g^−1^	Coating Binder Content	Area Density /mg cm^−2^	Cathode Sulfur Content/wt%	S Loading/mg cm^−2^	Capacity/(mAh g^−1^) (Rate)	Cycling Performance/(mAh g^−1^) (Cycles, Rate)	Capacity Decay/%	Ref.
Super P	-	-	None	0.2	55	1.1–1.3	1289 (0.5 C)	828 (200/0.2 C)	0.19	[26]
MWCNTs	2.76	410.42	None	0.17	55	-	1107 (0.5 C)	881 (150/0.2 C)	0.14	[27]
PG	3.361	1443	20 wt% PVP	0.54	63	1.8–2.0	1165 (0.5 C)	877 (150/0.5 C)	-	[28]
PC/MWCNT	0.17	83.4	10 wt% PVDF	0.51	70	1.6–1.7	911 (0.5 C)	659 (200/0.5 C)	0.138	[29]
GCFF	-	-	None	-	60	0.7	1280.14 (0.2 C)	1004.62 (100/0.2 C)	-	[30]
CFs	-	-	None	0.16	60	1	1280.14 (0.2 C)	683 (500/0.5 C)	0.071	[31]
rGO/CB	2.334	861.12	None	-	-	-	1014.5 (0.2 C)	850.9 (100/0.2 C)	0.17	[32]
CNT/AC	0.19	1312	20 wt% PVDF	-	70	-	1495.6 (0.2 C)	742 (200/0.2 C)	0.25	[33]
HCNF/rGO	-	-	None	1.2	60	1.4	1318.4 (0.2 C)	779.1(100/1 C)	0.13	[34]
mesoC	2.9	843	5 wt% super P and 10 wt% PVDF-HFP	0.5	70	3.5	1378 (0.2 C)	1021 (100/0.5 C)	0.081	[35]

## Data Availability

Data sharing not applicable.

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
