# Peer review of "A Review of the Application of Modified Separators in Inhibiting the “shuttle effect” of Lithium–Sulfur Batteries"

_membranes, 2022, doi:10.3390/membranes12080790_

Round 1

Reviewer 1 Report

The authors summarized different strategies for separator modification to improve the cycling life of Li-S batteries by preventing or minimizing the cross-over effects of LiPSs. I recommend publishing this review after minor revision.

1. The authors should discuss the relations between LiPSs absorption material modification on PP and the S loading in the cathode.

2. The authors should discuss whether the absorbed LiPSs can be released from the absorption materials, which determines the 1st cycle CE of Li-S batteries.

3. In general, when discussing electrochemical performance (like cycling performance), the authors should provide the S cathode formulation and loading, which are critical parameters people are interested in.

4. In the particle buildup section, could the author discuss the resistance change after adding nanoparticles to the separators? 

Author Response

Comments:

  1. The authors should discuss the relations between LiPSs absorption material modification on PP and the S loading in the cathode.

Response: When pp is modified with LiPSs absorption material, the S loading of the cathode can be increased to a certain extent, and the diffusion of LiPSs can be effectively suppressed. In addition, the diffusion coefficient of lithium ions does not decrease significantly [56].

[56] Chiu, L.L.; Chung, S.H. Composite gel-polymer electrolyte for high-loading polysulfide cathodes. J. Mater. Chem. A. 2022.

  1. The authors should discuss whether the absorbed LiPSs can be released from the absorption materials, which determines the 1st cycle CE of Li-S batteries.

Response: It can be known from several literatures we cited: the role of the modified layer of the LSBs separator is complex. Some can form a new combination with the separator through chemical interaction such as Li3PO4, thereby improving the lithium ion conductivity of the separator and promoting redox reaction of sulfur, but at the same time consumes some of the active sites of adsorbed LiPSs in the separator [53]. Some modified layers of separators, such as Fe3O4/RGO, whose unique structure can physically prevent the shuttle of LiPSs and form trap centers for dissolving LiPSs through strong chemical interaction with polysulfides. But since it is in close contact with the sulfur cathode and acts as an upper current collector, it causes some irreversible lithiation of the battery in the voltage range [48]. However, in general, these modified layers can improve the electrochemical performance of lithium-sulfur batteries. Therefore, further specific experiments may be required to discuss the desorption of LiPSs on the adsorbent material and its effect on the 1st cycle CE of LSBs only from the perspective.

[53] Wang, Z.; Feng, M.; Sun, H.; Li, G.R.; Fu, Q.; Li, H.B.; Liu, J.; Sun, L.Q.; Mauger, A.; Julien, A.M.; Xie, H.M.; Chen, Z.W. Con-structing metal-free and cost-effective multifunctional separator for high-performance lithium-sulfur batteries. Nano. Energy. 2019, 59, 390-398.

[48] Cheng, P.; Guo, P.Q.; Liu.; D.Q., Wang, Y.R.; Sun, K.; Zhao, Y.G.; He, D.Y. Fe3O4/RGO modified separators to suppress the shuttle effect for advanced lithium-sulfur batteries. J. Alloy. Compd. 2019, 784, 149-156.

  1. In general, when discussing electrochemical performance (like cycling performance), the authors should provide the S cathode formulation and loading, which are critical parameters people are interested in.

Response: Information on the S cathode formulation and loading has been added to the electrochemical performance discussion section of the manuscript.

  1. In the particle buildup section, could the author discuss the resistance change after adding nanoparticles to the separators? 

Response: The adding of nanoparticles to the separators can significantly reduce battery resistance by promoting charge transport at the cathode-separator interface, or by providing voids to accommodate various redox species and prevent solid lithium sulfide precipitation at the cathode-separator interface to form an insulating layer [64,65].

[64] Maletti, S.; Podetti, S.; Oswald, S.; Giebeler, L.; Barbero, C.A.; Balach, J. LiV3O8-Based Functional Separator Coating as Effective Polysulfide Mediator for Lithium–Sulfur Batteries. ACS Appl. Energy. Mater. 2020, 3, 2893-2899. 

[65] Ghazi, Z.A.; He, X.; Khattak, A.M.; Khan, N.A.; Liang, L.B.; Iqbal, A.; Wang, J.X.; Sin, H.; Li, L.S.; Tang, Z.Y. MoS2/Celgard Separator as Efficient Polysulfide Barrier for Long-Life Lithium-Sulfur Batteries. Adv. Mater. 2017, 29, 1606817.

Reviewer 2 Report

Please find the comments attached.

Author Response

Comments:

  1. 67 research papers are reviewed and summarized in this review article. The organization of the manuscript is good and easy to follow in each design concept and research case. The manuscript is well written. Some minor grammar errors and typos are suggested to be removed during the revision. For example, lines 31 and 35 should be S8 with a subscript 8; line 193 should correct the sample name of Ni3N and No2N. [Suggestion] Please review the manuscript and modify the tones.

Response: The errors mentioned in this comment have been corrected in the corresponding places in the article.

  1. Table 1 gives important information. It is suggested to provide the mass loading of the coating layer. It is also very helpful if the authors could give the fabrication information of with and without binder. [Suggestion] Please consider reporting the mass loading of the coating layer and the binder amount.

Response: The mass loading of the coating layer and the binder amount have been listed in Table 1.

  1. The review article seems to separate the modified separator as carbon coating and noncarbon coatings. However, according to the reference papers, many noncarbon coating papers have additional carbon to mix with other functional materials. This information is suggested to be mentioned and considered in this review. [Suggestion] Please consider reporting the use of carbon in the modified separators with other functional materials.

Response: The distinction between carbon materials and non-carbon materials occurs only in the physisorption part. The other parts are the synergistic modification of mixed materials of carbon and other functional materials.

  1. Some references summarize the materials in optimizing the performance of lithium-sulfur cells are suggested Molecules 2022, 27, 228 (various carbon coating materials) J. Mater. Chem. A 2022, 10, 13719-13726 (gel electrolyte coating) J. Power Source 2016, 334, 179-190 (carbon coating for long life) [Suggestion] Please support the discussion with reference citation and summarize the development trend.

Response: The references suggested in this comment have been cited in the article.

  1. In the conclusion section, it is suggested to give some general comments on the current research and the prospective on the future development for the lithium-sulfur cells with high energy density with suggested values and with lean electrolyte with suggested values. [Suggestion] Please consider giving comments and prospective in the conclusion section.

Response: The comments and prospective have been given in the conclusion section.

  1. Please review the references. The abbreviation of journal should all have a period after the abbreviated words.

Response: Periods after journal abbreviations have been added.

Reviewer 3 Report

Summary: The Review Article membranes-1855040 titled, “A review of the application of modified separators in inhibiting the "shuttle effect" of lithium-sulfur batteries,” reports a detailed summary of the key component in the lithium-sulfur batteries. A modified separator is designed to address the main issue of sulfur cathodes by blocking the free migration of dissolved polysulfides. Besides having the cell configuration in blocking the diffusion of polysulfides, the coating materials further offer the physical/chemical adsorption, catalytic adsorption, electrostatic effect, and multifunction in the optimization of the cell performance. General comment: This review paper makes a good summary of the research trend of the modified separator in terms of the research development and the material designs. Each milestone in the modified separators in lithium-sulfur batteries is supported with highlighted research articles. A minor revision is therefore suggested to provide some necessary information and parameters. Hope the authors feel the comment useful. Thank you. Comments: (1) 67 research papers are reviewed and summarized in this review article. The organization of the manuscript is good and easy to follow in each design concept and research case. The manuscript is well written. Some minor grammar errors and typos are suggested to be removed during the revision. For example, lines 31 and 35 should be S8 with a subscript 8; line 193 should correct the sample name of Ni3N and No2N. [Suggestion] Please review the manuscript and modify the tones. (2) Table 1 gives important information. It is suggested to provide the mass loading of the coating layer. It is also very helpful if the authors could give the fabrication information of with and without binder. [Suggestion] Please consider reporting the mass loading of the coating layer and the binder amount. (3) The review article seems to separate the modified separator as carbon coating and noncarbon coatings. However, according to the reference papers, many noncarbon coating papers have additional carbon to mix with other functional materials. This information is suggested to be mentioned and considered in this review. [Suggestion] Please consider reporting the use of carbon in the modified separators with other functional materials. (4) Some references summarize the materials in optimizing the performance of lithium-sulfur cells are suggested Molecules 2022, 27, 228 (various carbon coating materials) J. Mater. Chem. A 2022, 10, 13719-13726 (gel electrolyte coating) J. Power Source 2016, 334, 179-190 (carbon coating for long life) [Suggestion] Please support the discussion with reference citation and summarize the development trend. (5) In the conclusion section, it is suggested to give some general comments on the current research and the prospective on the future development for the lithium-sulfur cells with high energy density with suggested values and with lean electrolyte with suggested values. [Suggestion] Please consider giving comments and prospective in the conclusion section. (6) Please review the references. The abbreviation of journal should all have a period after the abbreviated words.

Author Response

Comments:

  1. 1. All the images cited from other journals should be licensed and authorized by publishers.  

Response: All the images cited from other journals are licensed and authorized by the publisher, as attached.

  1. “Doping functional particles in the novel separator can combine the steric hindrance 379 effect with other effects, enabling the separator to further inhibit the shuttle of LiPSs”. Elements doped methods are usually applied to improve the properties of materials. The following articles should be cited to support this work: Oxidative Degradation of Phenols and Substituted Phenols in the Water and Atmosphere: A Review, https://doi.org/10.1007/s42114-022-00435-0; Recent Advances in Co3O4as Anode Materials for High-Performance Lithium-Ion Batteries,

Engineered Science, 2020, 11, 19-30, https://dx.doi.org/10.30919/es8d1128  

Response: The references suggested in this comment have been cited in the article.

  1. Where’s the Figure 13b? I cannot find this image. The authors should check the manuscript carefully.   

Response: The icon was marked in the wrong order and has been corrected in the text.

  1. I would like the authors add some comments about the energy units like supercapacitor, Zlithium ion battery, et.al: Recent progress in cathode catalyst for nonaqueous lithium oxygen batteries: a review, https://doi.org/10.1007/s42114-022-00500-8; Recent advances in transition metal oxides with different dimensions as electrodes for high-performance supercapacitors: https://doi.org/10.1007/s42114-021-00358-2; Co3O4Nanoparticles Dotted Hierarchical-Assembled Carbon Nanosheet Frameworks Catalysts with Formation/Decomposition Mechanisms of Li2O2for Smart Lithium-Oxygen Batteries  

Response: The references suggested in this comment have been cited in the article.

  1. All the images in this work should be regrouped in this work, and more examples are suggested to be added to enrich this work.  

Response: The images in this work have been regrouped and more examples have been added to enrich this work.

  1. Prospect should be added in the last section of this review.  

Response: Prospects for the development of high-energy-density lithium-sulfur battery separators under high sulfur loading conditions have been added to the conclusion section.

  1. The manuscript contains spelling/grammatical errors. So, the language should be polished thoroughly.  

Response: We have carefully checked the spelling/grammatical errors of the article and made changes.